# Dorsal Anterior Cingulate Cortex Coordinates Contextual Mental Imagery for Single-Beat Manipulation during Rhythmic Sensorimotor Synchronization

**DOI:** 10.3390/brainsci14080757

**Published:** 2024-07-28

**Authors:** Maho Uemura, Yoshitada Katagiri, Emiko Imai, Yasuhiro Kawahara, Yoshitaka Otani, Tomoko Ichinose, Katsuhiko Kondo, Hisatomo Kowa

**Affiliations:** 1Department of Rehabilitation Science, Kobe University Graduate School of Health Sciences, Kobe 654-0142, Japan; ohtani@kobe-kiu.ac.jp (Y.O.); kowa@med.kobe-u.ac.jp (H.K.); 2School of Music, Mukogawa Women’s University, Nishinomiya 663-8558, Japan; ichino@mukogawa-u.ac.jp; 3Department of Bioengineering, School of Engineering, The University of Tokyo, Tokyo 113-8655, Japan; tkatagiri@g.ecc.u-tokyo.ac.jp; 4Department of Biophysics, Kobe University Graduate School of Health Sciences, Kobe 654-0142, Japan; e-imai@pearl.kobe-u.ac.jp; 5Department of Human life and Health Sciences, Division of Arts and Sciences, The Open University of Japan, Chiba 261-8586, Japan; kawahara2@ouj.ac.jp; 6Faculty of Rehabilitation, Kobe International University, Kobe 658-0032, Japan; 7Mitou Medical & Science Foundation, Tokyo 136-0062, Japan; kkfunky21@gmail.com

**Keywords:** dorsal anterior cingulate cortex, mental imagery, rhythm, sensorimotor synchronization, deep-brain activity

## Abstract

Flexible pulse-by-pulse regulation of sensorimotor synchronization is crucial for voluntarily showing rhythmic behaviors synchronously with external cueing; however, the underpinning neurophysiological mechanisms remain unclear. We hypothesized that the dorsal anterior cingulate cortex (dACC) plays a key role by coordinating both proactive and reactive motor outcomes based on contextual mental imagery. To test our hypothesis, a missing-oddball task in finger-tapping paradigms was conducted in 33 healthy young volunteers. The dynamic properties of the dACC were evaluated by event-related deep-brain activity (ER-DBA), supported by event-related potential (ERP) analysis and behavioral evaluation based on signal detection theory. We found that ER-DBA activation/deactivation reflected a strategic choice of motor control modality in accordance with mental imagery. Reverse ERP traces, as omission responses, confirmed that the imagery was contextual. We found that mental imagery was updated only by environmental changes via perceptual evidence and response-based abductive reasoning. Moreover, stable on-pulse tapping was achievable by maintaining proactive control while creating an imagery of syncopated rhythms from simple beat trains, whereas accuracy was degraded with frequent erroneous tapping for missing pulses. We conclude that the dACC voluntarily regulates rhythmic sensorimotor synchronization by utilizing contextual mental imagery based on experience and by creating novel rhythms.

## 1. Introduction

Rhythmic behaviors synchronized with external cueing are crucial for intrinsic human activities, such as creating music [1] and dancing [2]. How humans can extract rhythms from sound streams and achieve entrainment from rhythms, while rhythms are explicitly represented in the signals, is an interesting research question.

Rhythms with a wide variety of aspects, such as syncopation [3,4], are crucial for constructing attractive music. Such complex rhythms are based on flexible beat modification including beat-by-beat control [5]. Motor-related brain areas are required for detecting the beat embedded in such complex rhythms [6,7,8]. Numerous studies have investigated the neural mechanisms underpinning how rhythms are perceived and coordinated [9], typically including the neural resonance theory, which is based on two coupled oscillators at motor and auditory cortices and discussed in the frequency domain [10,11,12,13].

Beat perception is achievable through the capability of motor areas of predicting perceptual signals. To address this issue, a theoretical model [14,15,16] has been demonstrated [12,17,18,19,20] for generating predictive coding utilizing a neural network model including the premotor cortex and basal ganglia. A mathematical model associated with predictive coding based on Bayesian inference frameworks, where each pulse position is evaluated as a function of phase in neural oscillations, has been developed [21,22,23,24]. Such Bayesian approaches, however, suffer from issues associated with syncopation [25] (Figure 1), which could be crucial for complex rhythms [4]. This difficulty is attributed to the Bayesian framework per se, where the beat shift in syncopation is regarded as phase noise.

The current study focused on mental imagery to overcome the limitations of the Bayesian framework. Mental imagery is an internal model constructed based on memory without any sensory input [26]. By emulating the model, mental imagery not only contributes to the recognition of perceptual information, but also works as a predictor for upcoming events. We postulated that such mental imagery could be used for single-beat manipulation, even in the rhythmic sensorimotor synchronization [27,28].

First, we used a conventional neural mode to explain how mental imagery would work. As shown in Figure 2a, the medial prefrontal cortex (mPFC) and the hippocampus, involved in the default mode network, are activated [29,30] during imagery formation, suggesting that mental imagery is created in the mPFC and flexibly modified by self-referential processing [31], that is, divergent thinking [32,33]. Such flexible mental imagery generates a future scenario based on experience, which can be emulated [34] by the supplementary motor area (SMA), linked to the basal ganglia and involved in the cortico-basal ganglia-thalamo-cortical (CBGTC) network [35,36]. The SMA is involved in the motor network including also the primary motor (M1) and dorsal premotor (PMd) cortices, which share imagery information. The PMd is bidirectionally coupled with the auditory cortex via the inferior parietal lobule, inferior frontal gyrus (IFG), and the superior temporal gyrus (STG) [37], constructing a metacognitive control circuit [38]. This bidirectionality, indicating that imagery and perception share the same neural substrates [39,40], contributes not only to recognizing perceptual information, but also to selecting a desirable behavioral option by mental emulation [41]. Such cognitive performance can be degraded when the coupling is disrupted [42]. Accordingly, the SMA plays a crucial role in coordinating mental imagery [22].

Here, we propose an imagery-emulation framework for flexibly manipulating the beat (Figure 2b). For promptly engaging sequential events, responses are promoted in accordance with the mental imagery under appropriate inference before the events. Afterwards, the inference error between imagery and perception is evaluated to update mental imagery event-by-event to minimize the inference error. The dACC detects and evaluates the inference error from the metacognitive control circuit via the anterior insula cortex (aIC), involved in the salience network, further updating mental imagery. Hence, we hypothesized that the dACC, with versatile cognitive functions including reward-based goal-oriented decision-making [43,44,45,46], behavioral adaptation [47,48] via the SMA [49,50], and performance monitoring [51,52,53,54], could be critical in the proposed framework.

Recently, a repetitive transcranial magnetic stimulation (rTMS) method showed that distant brain regions including the dACC and the dorsolateral prefrontal cortex (DLPFC) can be co-activated via functional coupling [55]. Further, DLPFC activation via rTMS has been reported to improve both cognitive [56] and athletic performance [57,58], suggesting that the dACC and DLPFC could cooperatively contribute to fast but “intelligent” motor control, thereby supporting our hypothesis.

In this study, we aimed to show the validity of our hypothesis. To this end, we employed a missing-oddball task [11,12] conducted using conventional finger-tapping paradigms [59,60], where participants were asked to precisely make relevant responses to real pulses while inhibiting irrelevant responses to missing pulses. Motor outcomes are required to satisfy such task conditions, and thereby, motor control modality is highly relevant when discussing this issue.

There are two modes of motor control: proactive and reactive. The proactive control mode is promoted based on predictive mechanisms in cognitive processing [61] for modulating upcoming action inhibition or adaptation consistent with environments [61,62,63]. During proactive control, a specific motor effector is anticipatorily selected while maintaining goal-relevant information in the DLPFC to manifest optimal behavioral performance [64]. Such proactive motor control is supported by pre-SMA and IFG for executing intended motor outcomes, including potential covert imagery motor acts [62]. In contrast, the reactive control mode is characterized by stimulus-driven responses adapted by the late correction mechanism [64,65,66] via goal reactivation. Wider brain networks involving the lateral prefrontal cortex, which are associated with episodic memory [67], contribute to this reactive control mode [64]. Interestingly, SMA contributes to prompt adjustment of behaviors consistent with external stimuli [68,69]. This means that SMA is crucial in both proactive and reactive motor control modes [70]. However, the dACC, positioned upstream of the SMA [49,50,71], takes the initiative in motor control, including rapid stopping via a hyper-direct pathway activated by the dACC-subthalamic nuclei connection [72,73]. The two proactive and reactive motor control modes are independent [74] but simultaneously activated [75]. This enables flexible switching between these modes for achieving higher control performance [76,77].

Based on the proactive/reactive control framework as mentioned above, further consideration was given to our specific motor control modality issue during the missing-oddball task. To begin with, the cued-tapping manner during the missing-oddball task would be determined in accordance with the imageries. We postulated that predictive tapping would be manifested as real pulses in the regular portions of the missing-pulse sequence under proactive-tapping imagery. Such imagery-driven proactive tapping is accompanied by negative mean asynchrony (NMA) to achieve on-pulse tapping performance based on coincident detection between auditory and tactile signals via sound and tapping [78,79,80] (Figure 3a). This tapping manner will necessarily result in unavoidable erroneous tapping for missing pulses (Figure 3c). In contrast, reactive-tapping imagery involves stimulus-driven motor processing schemes (Figure 3b). If the reactive tapping is maintained in the regular portions under reactive-tapping imagery, erroneous tapping for missing pulses will be successfully avoided (Figure 3d).

Considering that proactive and predictive motor control modes can co-exist [64,81,82,83], we further predicted the case that proactive- and reactive-tapping imageries will be flexibly exchanged according to real or missing stimulus condition. Such flexible imagery exchange would resolve the conflict associated with the speed–accuracy tradeoff [84,85,86]. The cued-tapping task synchronously with the missing-pulse sequence provides two conflicting requirements. Proactive-tap imagery provides on-tap performance with small asynchrony but has a risk of erroneous tapping for the missing pulses. In contrast, the reactive-tap imagery provides accurate tap performance by avoiding erroneous tapping, but suffers a response delay with large asynchrony. Accordingly, the proactive- and reactive-tap imageries are conflicting [84,85,86]. Therefore, the flexible imagery exchange framework is considered suitable for resolving this conflict, as shown in Figure 3e. If our hypothesis was valid, cued tapping should be consistent with the flexible imagery exchange, indicating involvement of the dACC. Hence, we measured brain responses during two tapping tasks (Task 1 and 2 for regular and missing-pulse sequences, respectively), utilizing the event-related deep-brain activity (ER-DBA) method previously developed for evaluating dACC dynamics [87]. In addition, conventional event-related potential (ERP) data served to determine which imagery would be selected by analyzing omission responses at the missing sites [11,12,88], and performance data were used to evaluate cognitive bias in decision-making during explainable imagery selection based on signal detection.

## 2. Materials and Methods

### 2.1. Stimuli and Experimental Procedure

Figure 4 shows an outline of the experiments conducted in the current study. We prepared two pulse sequences. One, a regular sequence with 120 pulses having an equal interval of 1000 ms for Task 1 as a control task (Figure 4a), and another, a missing-pulse sequence with 45 missing and 255 real pulses with the same interval as in the regular sequence, constructed by randomly omitting pulses from a regular sequence for Task 2 as in the missing-oddball task (Figure 4b). The sequence structure was specified by a numbering scheme whereby real pulses were labeled by the lag number such as −2, −1 (for prior pulses), 1, 2, ▪▪▪ (for posterior pulses) from each missing pulse (N = 0). The sequence was also segmented by missing pulses, providing regular portions characterized by the number (M) of involved real pulses.

Participants had to wear an EEG cap and were seated facing a desk with a laptop computer (Figure 4c). The computer presented auditory stimuli according to the provided sequence. All participants kept their eyes closed while executing the cued-tapping tasks. Further, they were asked not to move their heads or grind their teeth to suppress artifacts. Participants were asked to press a computer key along with stimulus presentation. The missing-pulse task was followed by a conventional GO/NOGO paradigm in which prompt keystrokes for real pulses were required. No strokes for missing pulses were considered as erroneous tapping, and no keystrokes for real pulses were considered omission errors. All participants performed Task 1 first, followed by Task 2. The time necessary for completing both tasks was ≤10 min including an intermission time of a few minutes. The EEG cap had preassembled Ag/AgCl electrodes consistent with the international 10–20 method, including the mastoid electrodes (A1 and A2) as reference (Figure 4d). EEG signals from the cap were recorded by a digital (24 bit) EEG system (Nexus 32, Mind Media BV, Roermond-Herten, the Netherlands) at a sampling rate of 256 Hz. The EEG system was regulated by control software (Bio Trace+ Software for Nexus 32 Version: V2009a4) installed in the computer. The system regulated sequential stimulus presentations and integrated EEG data with the stimulus onset and tap times recorded in an event channel with a sampling rate of 512 Hz. To avoid electrical noise, a wireless keyboard was placed on a conductive sheet linked to the participants via an electrical cable to suppress static electricity; this was used to maintain complete electrical isolation of the EEG system, which was similarly linked to the computer via optical fiber cables.

We started by determining an appropriate participant number for satisfying statistical requirements with an appropriate sample size. Taking into account the fact that EEG signals were primary measures in the study, we estimated the sample size based on the statistical evaluation via pre-experimental results associated with EEG measures, including the DBA index as mentioned later. The typical sample trace (Figure 4e) depicted a sinusoidal curve during isochronous cued tapping, providing an excursion amplitude $\Delta =1.95\;\mu V^{2}$ and pooled standard deviation $SD=5.53\;\mu V^{2}$ taking into account standard deviations at the peak (*SD1*) and bottom (*SD2*) $\left(SD=\surd (\frac{SD1^{2}+SD2^{2}}{2})\right)$. Assuming that statistical significance might be required for a difference of around Δ/3, the necessary minimal sample size was estimated under an assumed significance level of 0.05 as
$$N={\left(\frac{1.96SD}{∆/3}\right)}^{2}=279$$

We conducted a missing-oddball task including M missing pulses. The task provided two behavioral manners for the missing pulses: erroneous tapping and successful avoidance. Assuming an error rate of 0.3 based on pre-study experiments, an expected number was around 13 assuming the missing pulse number of 45. Hence, the minimal necessary participant number was estimated at 279/19 ≒ 21.

### 2.2. Participants

Taking into account the minimal necessary participant number as estimated in the previous session and an estimated exclusion rate of around 0.3, we determined the minimal participant number of 30. Further considering a risk of incidence during experiments, we recruited 33 healthy (without daily use of psychotropic drugs) student volunteers including 17 females (mean age, 22.0 ± 3.6 years) from Mukogawa Women’s University, Kobe International University, Kobe Co-medical College, Tokyo University of Foreign Studies and Kanagawa University, Japan.

### 2.3. Data Analysis

#### 2.3.1. Behavioral Performance

Behavioral performance was quantified by reaction times (RTs) and tap accuracy. RTs were evaluated from pairs of event-onset and key-stroke markers in every epoch, ranging from −500 to 500 ms, and time-locked to the event onset recorded in the event channel at a sampling rate of 512 Hz. To examine whether cued tapping could be promoted according to the proactive and reactive motor control modes (Figure 3a,b), we evaluated pooled RT histograms across all participants for the three conditions, i.e., real pulses in the regular sequence (Task 1) and real/missing pulses in the missing-pulse sequence (Task 2) (Figure 5a). After confirming the co-existence of proactive and reactive tapping for the missing-pulse sequence (Task 2), represented by two separated populations in the pooled histogram, we examined individual behavioral performance by evaluating RT histograms as a lag-number (N) function locked to the missing site (N = 0). Next, histograms were analyzed using the kernel density estimation method [89] to derive peak RTs and the ratio between the two populations from dual-lobe distribution functions.

Tap accuracy was evaluated for each participant by counting the erroneous taps for missing pulses. Considering the individual cognitive processing ability, we evaluated the error rate as reverse accuracy as a function of the corresponding RT for the erroneous tap. The error rate versus RT was analyzed by conventional regression analysis.

To examine whether tap accuracy could be attributed to decision criteria or perceptual bias, we analyzed the behavioral performance based on SDT [90,91]. To apply SDT to the missing-oddball task, we started by categorizing behavioral responses into four types, depending on mental imagery. For the proactive-tap imagery, tapping for real pulses is predictive before pulse perception and regarded as a false hit (FH), while tapping for missing pulses is erroneous and regarded as a FA (Figure 5b). In contrast, for the reactive-tapping imagery, tapping for real pulses is triggered by perception and regarded as a CH while tapping for the missing is postponed, i.e., avoided as a correct rejection (CR) because no signals could initiate execution (Figure 5c). Figure 5d shows our modification to SDT. We postulated two different probability density functions for real $\left(P_{R}\left(x\right)\right)$ and missing pulses ($P_{M}\left(x\right))$ as a function of decision variable x, defined as an indicator directed toward answering “NO” which means “Pulses are absent.” We further defined c as an indicator of decision bias, indicating that tapping is inhibited for x > c while tapping is promoted for x < c. Hence, we could readily evaluate the correct-hit ratio (CHR) and correct-rejection ratio (CRR) utilizing the distribution functions:$$CHR=\int_{c}^{\infty }P_{R}\left(x\right)dx$$
$$CRR=\int_{c}^{\infty }P_{M}\left(x\right)dx$$

Here, false-hit ratio (FHR) and false-alarm ratio (FAR) are not independent parameters as FHR = 1 − CHR and FAR = 1 − CRR.

According to SDT, we defined the SDT indices d-prime and criterion c as
$$d^{\prime }=Z\left[CHR\right]-Z\left[CRR\right]$$
$$c=-\frac{1}{2}\left\{Z\left[CHR\right]+Z\left[CRR\right]\right\}$$
where Z indicates the inverse of the cumulative normal distribution. CRR was derived from the rate of erroneous taps for missing pulses and CHR from the ratio between reactive and proactive taps for real pulses in the missing-pulse sequence as
$$CHR=\frac{P_{C}}{P_{C}+{P_{F}}^{\prime }}$$
where P_F_ and P_C_ are the population heights of false hits and CHs, respectively. d-prime represents the sensitivity for distinguishing real and missing pulses, while the criterion c represents cognitive bias, such that tapping is more inhibitory (conservative) for c < 0 but stimulatory for c > 0.

Hence, we examined the change in d-prime and c as a function of lag number (N) locked to the missing-pulse site based on regression analysis.

#### 2.3.2. Event-Related Deep-Brain Activity Method

The ER-DBA method was developed for evaluating dynamic properties of the dACC based on a previous study [92], revealing that the occipital EEG alpha 2 (10−13 Hz) power reflects activities of the ventral tegmental area and dACC by lower (<0.04 Hz) and higher (>0.1 Hz) fluctuation components, respectively. We previously demonstrated the validity of the ER-DBA method [87] to evaluate the role of the dACC in cognitive processing, including performance monitoring and decision-making for motor execution. The study has shown that the ER-DBA method is consistent with the conventional alpha event-related (de)synchronization (ERD/ERS) methods [93,94,95,96]. Before applying the ER-DBA to analyze the brain responses during the missing-oddball task, we confirmed the validity of the method in similar cued-tapping tasks. Hence, we conducted two cued finger-tapping tasks utilizing regular and random sequences in 14 healthy young adults. In line with previous research [59], a regular sequence promotes tapping with NMA while a random sequence promotes positive mean asynchrony. We obtained similar behavioral performance by pooled histograms of RT for the two tasks (Figure 6a). We further evaluated grand-averaged ER-DBA traces normalized by the median of the upper and lower trace envelopes (Figure 6b). For the regular sequence, the ER-DBA trace depicted deactivation and exhibited dissolution corresponding to tap timing at around −50 ms. In contrast, for the random sequence, the trace showed activation until tap timing but exhibited a mini dip and disappeared corresponding to the tap timing at around 200 ms. Since the negative and positive asynchronies associated with tapping are attributed to proactive and reactive motor control modes, aspects of the ER-DBA traces as mentioned above are considered to reflect the features of the motor control modality. While the ER-DBA traces depicted different aspects of activation or deactivation in accordance with sequence conditions, dip disappearance corresponded to tap timing. Based on these prior experiments, we evaluated motor control modality (proactive or reactive) by utilizing the ER-DBA trace features.

The implementation of the ER-DBA method was as follows. Occipital EEG amplitude signals from O1 and O2 electrodes (256 SPS (sample per second)), referring to the mastoid electrodes (A1 and A2), were filtered (third-order Butterworth digital filter) to extract the alpha2-band (10−13 Hz) component and smoothed using a moving average (data point number = 8). Hence, the two 32 SPS time-series EEG alpha-2 amplitude signals for O1 and O2 were integrated to calculate the DBA index defined as the mean actual power of these amplitude data.

Data for each trial were extracted in epochs ranging from −1000 to 8000 ms around each pulse/missing-pulse onset marker from the time-series DBA data. Deviant trials were firstly excluded by mean of visual inspection. Then, the mean and standard deviation of the trial data were evaluated for excluding deviant trial data as artifacts. The exclusion criterion was set at three times the standard deviation. The trial data were categorized according to four kinds of stimulus-onset markers consistent with four behavioral performances as hit (tap for real pulses), missing (no response for real pulses), correct rejection (relevant avoidance for missing pulses), or false alarm (irrelevant tap for missing pulses). Then, all trial data were grand averaged over all participants for every category to derive the grand-averaged ER-DBA traces time-locked to stimulus onset. The baseline correction was conducted in the range from −500 to 0 ms for each grand-averaged ER-DBA trace. The trace accuracy was evaluated using the standard error of the mean (SEM).

#### 2.3.3. ERP Trace Analysis

Since the cued-tapping tasks utilized auditory pulse presentations, the ERP traces were expected to provide typical late auditory evoked potentials (LAEPs) including N100, P200, N400, and the late positive complex (LPC) [97]. The early components of N100 and P200 as exogenous ERPs reflect sensory gating [98], while the late ERP components of N400 and LPC as endogenous ERPs reflect higher-order cognitive processing [99,100,101]. In contrast, the ERP traces for the missing pulses were expected to represent reverse traces for the real pulses. Such reverse ERPs were regarded as omission responses, reflecting comparison between imagery and perception [88,102,103,104,105,106]. Accordingly, we visually inspected ERP waveforms in defined time windows to detect N100, P200, N400, and LPC for real pulses and their reverse for the missing pulses. The EEG signal at the vertex electrode (Cz) was pre-processed using a digital bandpass filter (third-order Butterworth digital filter) with a 0.01–45 Hz window.

The digital time-series data were similarly extracted in the range of −1000 to 8000 ms around the stimulus-onset markers. Deviant trials with extraordinary large amplitudes as artifacts were excluded via visual inspection. Then, the mean and standard deviation of the trial data were calculated to exclude deviant trials. The exclusion criterion was set at three times the standard deviation. The trial data were similarly categorized to the four performance groups. Then, the trial data were grand averaged for each category to evaluate grand-averaged ERP traces time-locked to the stimulus (real or missing pulses) onset. The baseline correction was conducted for the grand-averaged ERP traces in the range from −200 to 0 ms.

#### 2.3.4. Statistics

Individual mean RTs were statistically evaluated by SEM. Comparison of individual RTs between trials with different lag numbers during Task 2 (the missing-oddball task) was conducted using ordinary one-way ANOVA and the Holm–Sidak t-test multiple comparisons test. The significant level was set at *p* = 0.05. Sample size was 33. The correlations of FAR versus mean RT of erroneous taps, FHR versus FAR, FAR versus d-prime, and FAR versus criterion (c) were evaluated by the Pearson’s coefficient of correlation (r) and ANOVA indices including F-values, *p*-values, and effect size (η^2^). The sample size corresponded to the number of participants providing effective data (33). Significance was set at *p* = 0.05. Multiple comparison of CHR was conducted using one-way ANOVA and the Holm–Sidak t-test. The significant level was set at *p* = 0.05 and sample size was 33.

Grand-averaged ER-DBA and ERP traces were evaluated by SEM bands. The sample number was up to 1485 (45 × 33) for missing pulses and 8415 (255 × 33) for real pulses; this number decreased by excluding artifact and electric noises. Comparison of the ER-DBA averaged in the epoch of interest between different conditions was conducted by the t-test with significance set at 0.05. The different conditions included behavioral differences between negative and positive asynchronies of taps for real pulses in the missing-pulse sequence. We also utilized a t-test for comparing averaged DBA index levels during taps for real pulses between tasks. Calculations were conducted utilizing a commercially available statistics software installed in the Origin Pro 2022b, OriginLab Corporation, Northampton, MA, USA.

## 3. Results

### 3.1. Behavioral Performance

We investigated pooled RT histograms of taps during tasks for all participants. Taps for real pulses of the regular sequence (Task 1) depicted NMA centered at around −50 ms, while a small population had positive mean asynchrony at around 200 ms (Figure 7a). Taps for real pulses in the missing sequence (Task 2) depicted dual lobes consisting of populations with negative and positive asynchronies. Their peaks were observed at around −50 for the negative and 200 ms for the positive asynchrony population, consistent with the histogram in Task 1 (Figure 7b). Taps for missing pulses in Task 2 provided a single-lobe population of the RT histogram (Figure 7c), whose center was located at the event onset (Time 0). These results indicated that cued tapping could be promoted by proactive and reactive control modes. To explore how dual motor control contributes to regulating behavioral performance for the missing pulses regarded as salient events, we investigated individualized mean RTs as a function of lag number (N) locked to the missing pulse site (N = 0) (Figure 8a). While the individual mean RTs were distributed from negative to positive temporal regions, the overall mean RT depicted characteristic trajectories as a function of lag number, with the mean RT minimized at the missing site (N = 0), then maximized at N = 1, and much decreased at N = 2 (Holm–Sidak *p* < 0.0001) and 3 (Holm–Sidak *p* < 0.0001), gradually increasing with the lag number (See Table 1 for numerical data). Whereas such complex behavioral performance closely resembled transient responses, the strong lag dependence of the tap performance was attributed to the population balance between the proactive and reactive motor control modes (Figure 8b). This was supported by the individualized histograms of the CH ratio, whose trajectory as a function of lag number traced the similar aspect of the overall mean RT while maintaining almost constant peak RTs of the dual populations (Figure 8c).

The population balance between proactive and reactive taps is crucial for understanding how imagery contributes to regulating tapping for missing pulses. However, it was difficult to evaluate this balance because the missing-pulse site did not promote CH responses. To compensate for this, we investigated the correlation between the error rate (FAR) and the reaction time of erroneous taps. We expected that FA would be inhibited in the positive time zone (t > 0) when real or missing pulses are determined, while no inhibitory signals would exist in the negative time zone (t < 0) when real or missing pulses are completely uncertain. This suggested that FAR would decrease with increasing RT of taps for missing pulses. Figure 9 shows a significant (r = −0.737, F = 33.4, η^2^ = 0.544, *p* = 3.32 × 10^−6^) negative correlation between FAR and reaction time.

The previous results hinted at the crucial role of shifting the balance between proactive and reactive control modes in regulating tapping for missing pulses. Hence, using the SDT, we examined whether this shift would be attributed to mental-imagery-promoted cognitive bias or perceptual bias. We evaluated as a function of the lag number (N) both d-prime as detectability (Figure 10a) and c as cognitive bias (Figure 10b). d-prime was maximized at around the missing-pulse site while c was minimized. d-prime gradually increased with increasing lag number, while c gradually decreased. The maximal d-prime at the post-missing site (N = 1) manifested statistical significance compared with those of the other sites based on the Holm–Sidak multiple comparisons test. The minimal criterion at N = 1 manifested a reverse aspect of d-prime, but was not significant compared with those with the other sites (See Figure 10a and Table 2 for numerical data). This coordinated change of d-prime and c implied that the cognitive and perceptual bias may not be exclusive. Hence, we examined the correlation between FHR and FAR for the trial just prior to the missing-pulse site (corresponding lag number N = −1; Figure 10c). The FHR versus FAR exhibited a significant (Pearson’s coefficient of correlation r = 0.795, F = 36.2, *p* = 5.6 × 10^−6^, η^2^ = 0.63) positive correlation, providing a cut-off FAR at around 0.36, where FHR became zero. This critical FAR corresponds to a reaction time of around 100 ms, indicating a lack of FHR with RTs >100 ms. Considering FHR + CHR = 1, such replacement may improve detectability at around the missing-pulse site. Whereas both d-prime and c were coordinated, c was more crucial for regulating the tapping manner for missing pulses since c showed a significant positive correlation with FAR (r = 0.86, F = 80.7, *p* = 1.3 × 10^−9^, η^2^ = 0.75; Figure 10d) while d-prime did not (r = 0.41, F = 6.25, *p* = 0.0178, η^2^ = 0.167; Figure 10e).

### 3.2. ER-DBA Trace Analysis

Taps for real pulses provided similar grand-averaged ER-DBA traces independently of sequence differences (regular or missing; Figure 11a), while the average ER-DBA index significantly differed (*p* = 0.00037, power = 0.94), suggesting missing pulses may require more cognitive effort than a simple case with no salient events like missing pulses for which erroneous tapping must be avoided. Whereas ER-DBA traces depicted similar sinusoidal excursions synchronous with external isochronous cuing at every 1000 ms, there was a slight reaction time difference, which provided different dissolution times of ER-DBA trace dips. To explore the discrepancy between the features of ER-DBA traces for the same stimulus condition (stimuli by real pulses), we checked ER-DBA traces during tapping for real pulses in the missing-pulse sequence (Task 2) by categorizing ER-DBA responses to two groups with negative and positive RT. Figure 11b shows the ER-DBA traces of the two groups, indicating that the negative-reaction-time group exhibited deactivation (ERD) while the positive-reaction-time group exhibited activation (ERS). Such an ERD/ERS decision was confirmed by a significant power difference (*p* = 0.036, power = 0.55) in the epoch marked by a gray area. The two groups similarly showed ER-DBA dips in between neighboring pulses for both negative and positive responses. These dips disappeared in accordance with each tap timing accompanied by negative/positive asynchrony. We further investigated responses for missing pulses in ER-DBA traces. Figure 11c shows differences in ER-DBA traces between avoidance and erroneous tapping for missing pulses time-locked to the missing-pulse onset. For avoidance of erroneous tapping, the ER-DBA trace exhibited activation, with no dips. In contrast, for erroneous tapping, the trace exhibited deactivation with a dip at around the reaction time at Time 0. These results imply that reactive motor control could avoid erroneous tapping for missing pulses, while proactive motor control could promote erroneous tapping. Next, we investigated the post-missing ER-DBA traces. Independently of the tapping manner at the missing site, ER-DBA traces exhibited activation (ERS) accompanied by peaks (marked by pink triangles) in the positive time zone corresponding to RT with strong positive asynchrony in the post-missing trial (lag number N = 1). However, for the next (lag number N = 2) trial, ER-DBA traces exhibited deactivation (ERD) accompanied by dips (marked by blue triangles) in the negative time zone corresponding to RT shifting toward negative asynchronies. We then investigated post-missing temporal development of ER-DBA traces (Figure 11d). We found that the proactive motor control mode, characterized by ER-DBA dips (marked by orange arrows) in the negative time zone, was dominant in the early stage of the temporal development of N (lag number) < 3, while the reactive mode characterized by mini-dips (marked by green arrows) in the positive time zone became dominant in the late stage of N > 3. The gradual ER-DBA excursion, evaluated as the median of the upper and lower envelopes of ER-DBA traces, depicted a gradually changing trajectory (red line) maximized at around the onset of the post-missing trial.

### 3.3. ERP Trace Analysis

We identified typical LAEPs including P1, N1, P2, N2, and P3 (LPC) for real pulses, independently of regular or missing-pulse sequences (Figure 12a). There were no significant differences between regular and missing-pulse sequences except for the ERP N2 component. The N2 component in the regular sequence showed a smaller latency of around 300 ms, while for the missing-pulse sequence it had a larger latency, of around 400 ms. This suggested that N2 should be N300 for the regular sequence but N400 for the missing-pulse sequence [107]. In contrast, we found significant differences in ERP traces for the missing pulses. As shown in Figure 12b, the trace of erroneous taps as FA represented a pattern involving oN1 and oP2. The trace was almost a reverse of the ERP trace for real pulses in Task1, because oN1 and oP2 corresponded to P2 and N400 of the ERP trace for the real pulses, whereas the ERP component corresponding to the N1 (N100) was not present.

In contrast, the ERP trace of avoid regarded as CR only showed oP2, while missing showed oN1. We further found that erroneous taps with negative reaction times (RT < 0) more clearly depicted a reverse pattern of ERP traces for taps for real pulses, compared with that for taps with positive reaction times (RT > 0; Figure 12c). We finally investigated post-missing temporal development of ERP traces (Figure 12d). The ERP trace in the early stage of temporal development at N (lag number) = 2 did not differ from that for taps in Task 1 but gradually improved the contingent negative variation (marked by green color) and amplitude of the N100-P200 complex in the late stage.

## 4. Discussion

We evaluated behavioral performance during cued-tapping tasks with regular and missing-pulse sequences using the ER-DBA method for evaluating the dACC’s dynamic activity, as supported by SDT and ERP analyses. We demonstrated the validity of the hypothesis that the dACC can coordinate imagery-driven pulse-by-pulse rhythmic sensorimotor synchronization. The key findings were associated with contextual mental imagery [107,108], as follows.

### 4.1. The DACC Utilizes Contextual Imagery for Dynamic Regulation of Cued-Tapping Mode

Two conventional types of tapping characterized by negative and positive mean asynchronies [109] simultaneously appeared in the same missing-pulse sequence (Task 2); similarly, the mode in the regular sequence (Task 1) was in line with previous studies [110,111,112]. Such dual-tapping modes are attributed to different neural mechanisms of motor control, proactive and reactive, respectively [60]. Hence, the simultaneous tapping modes differently accompanied by deactivation and activation of the ER-DBA trace suggest that the dACC might flexibly regulate the dual control framework based on the context in accordance with the missing-pulse sequence.

Further, erroneous tapping for missing pulses was avoidable when the ER-DBA trace showed activation but unavoidable when it showed deactivation. As previously reported [113], the alpha ERS, defined as the activation as the EEG alpha power increase, represents functional inhibition. This indicates that ER-DBA activation reflects functional inhibition of the dACC. The reactive control of cued tapping requires inhibition of unnecessary imagery-driven outcomes via the dACC-SMA pathway until the appearance of appropriate external signals for guiding behavior to task goals [114]. Thus, it is reasonable that the reactive control associated with ER-DBA activation reflects dACC inhibition, which contributes to avoidance of erroneous tapping for missing pulses. In contrast, ER-DBA deactivation represents dACC excitation, typically involving selective inhibition for goal-directed cognitive control [115] based on self-referential processing [116] rather than external events [117]. Accordingly, the imagery-driven proactive control characterized by ER-DBA deactivation may cause erroneous tapping for missing pulses.

Moreover, the omission responses, ERP traces specific to the tapping mode, indicated that erroneous tapping was associated with oN1 and oP2, while successful avoidance of erroneous taps was associated with only oP2, suggesting that ER-DBA deactivation and activation are attributed to real and missing pulses, respectively. Notably, since oP2 is internally evoked during cognitive processing depending on context, the imagery might involve stimulus-response experiences associated with finger tapping. Overall, we confirmed that the dACC regulates rhythmic sensorimotor synchronization in accordance with contextual mental imagery via proactive and reactive motor control.

### 4.2. Maintenance of Mental Imagery

Tapping modes manifested pulse-by-pulse changes around the missing-pulse trials. As mentioned in the Introduction, we predicted that such changes could be promoted by strategic selection of mental imagery for the optimal tapping mode to overcome the speed–accuracy tradeoff. Hence, we examined the mental imagery changes in detail.

Figure 13a summarizes our results. The behavioral performance suggests the existence of two behaviors. One starts at the trial prior (N = −1) to the missing site (N = 0) by showing CH under imagery that was “missing,” resulting in successful avoidance of erroneous tapping as CR at the missing-pulse site under the maintained “missing” imagery. Such imagery was maintained until the next trial (N = 1), showing reactive tapping with CH. However, since the imagery changed from “missing” to “real”, proactive tapping (FH accompanied by NMA) was manifested at the next to post-missing trial (N = 2). In contrast, the other behavior started with FH at the prior trial (N = −1) under “real” imagery, resulting in erroneous tapping for missing pulses at the missing site (N = 0) under the same “missing” imagery. In this case, the imagery was updated from “real” to “missing”.

Importantly, imagery updating was promoted for both behaviors in accordance with event changes, such as from “missing” to “real” (N: −1 → 0) or “real” to “missing” (N: 0 → 1), without inference error. Whereas such imagery updating scheme is valid at around the missing-pulse trials, we observed tapping mode changes for higher lag numbers (N ≥ 4) such that the reactive control mode characterized by ER-DBA activation becomes more dominant with increasing taps with positive asynchrony, suggesting that the mental imagery associated with “missing” becomes stronger, despite real pulses being continuously presented. Moreover, such behavioral changes were asymmetric, considering that the FH frequency with RT > 100 ms selectively decreased.

Initially, we considered that the Bayesian brain [117,118,119,120] could explain such imagery changes, because the frequency of the “missing” imagery could be explained by the posterior probability distribution of encountering the missing pulses increasing at every hit of the real pulses. However, this probabilistic model cannot explain why and how only FHs with larger RT were excluded. The Bayesian model was also unable to explain the post-missing reactive responses characterized by positive mean asynchrony reflecting the “missing” imagery. Since almost all missing-pulse trials were embedded alone in the sequence, the imagery for the post-missing trial should be “real”, invalidating the Bayesian inference. Hence, based on our behavioral results, we consider that imagery changes could be promoted by abductive reasoning [121,122].

Proactive and reactive control can be simultaneous [75]. Whereas proactive responses manifest NMA, the Gaussian distribution ranges from negative to positive times. Thus, delayed proactive responses may be cancelled by the responses, characterized by mean positive asynchrony >100 ms (Figure 13b). Consequently, the contextual experience consists of reactive responses, promoting abductive reasoning that the reactive responses should be attributed to the imagery of “missing,” while the original imagery is “real,” thus shifting the cognitive bias from “real” to “missing” (Figure 13c). Based on the above, we developed a neural network model (Figure 13d). The auditory signal generated by real pulses is transmitted to the PMd for generating motor output from the primary motor cortex as M1. This is supported by the event coding theory [123]. In contrast, the IFG involved in this dorsal pathway sends an alarm signal to the dACC via the aIC [124,125]. Accordingly, the dACC promotes the pre-SMA to cancel the proceeding proactive responses by utilizing the hyper-direct pathway linked between the pre-SMA and subthalamic nucleus [126,127,128] as the interface of the CBGTC network. Our results provide no direct evidence for showing the imagery changes by abductive reasoning. However, the increasing population of the reactive tapping with increasing lag number from the missing-pulse site supports such irreversible imagery change.

### 4.3. Is Modal Completion Caused by Fusion of Imagery and Perception?

Under proactive motor control, there was highly frequent tapping for missing pulses accompanied by stronger NMA. Such tapping was characterized by clear omission responses compared with those for the tapping with rather positive RT, suggesting that vivid imagery may promote frequent tapping. Considering task regulation, tapping for the missing pulses was erroneous. However, considering that the missing-pulse sequence was imperfect, tapping was considered a normal activity for reintegrating the missing-pulse sequence to a regular one. Such filling-in-the-blank tapping [129,130] could be associated with modal/amodal completion [131,132]. In addition, the auditory illusion [133,134,135] generated by modal completion might promote tapping for missing pulses. However, since tapping was executed under negative asynchrony while illusion as modal completion could be attributed to coupling or fusion between imagery and perception [136] promoted in the positive time zone, the above postulation contradicts causality.

We therefore consider that a simpler neurophysiological mechanism may promote such filling-in-the-blank tapping, as indicated by the ER-DBA trace with dual dips around the midpoint between two neighboring pulses in regular portions of the missing-pulse sequence (Task 2), suggesting that a syncopated rhythm characterized by bootstrap structures [4,137] was constructed from the beat sequence. As previously reported, syncopation promotes motor movements [3,138,139]. Hence, the ER-DBA results suggest that mental imagery may produce a syncopated rhythm from the missing-pulse sequence for cued tapping independent of negative or positive asynchrony. To explain how the syncopation imagery differently affected the two tapping modes, we consider that the bootstrap might be tightly bound during “real pulse” imagery, maintaining stable repetitive tapping against the disturbance imposed by the missing pulses, while the strap might be so loose that tapping reactively accorded with external cuing under proactive inhibition. Accordingly, ER-DBA activation and deactivation may reflect loose and tight bootstrap binding in the syncopation imagery, respectively.

### 4.4. Limitation and Perspective

We have discussed frequent erroneous tapping with dominant negative asynchrony for missing pulses, which could be promoted by strong vivid imagery of bootstrapped syncopation rhythms. We consider the clear omission response as evidence supporting the above claim. However, we question whether omission responses can really represent mental imagery. This is because the timing dissociation between the imagery-driven proactive tapping, accompanied by strong MNA and omission responses having almost the same onset timing as that of the ERP response for real pulses, cannot be unexplainable if omission responses are tightly coupled with imagery. Thereby, we reconsidered interpretations of the omission responses, whereas many studies [105,140,141,142] based on the reverse hierarchy theory of imagery [143] have utilized omission responses for evaluating top-down cognitive processing by eliminating perceptual processing inadequately affected by confounding exogeneous responses [12,144].

Our results show clear differences in omission responses that erroneous tapping provided, i.e., oN1, oP2, and oP3, while successful avoidance only provided oP2 and oP3 with missing oN1. oN1 corresponds to N200 as a neural marker of response inhibition [145,146,147]. This means that inhibitory activity could be proactively programmed under “real pulse” imagery and triggered as soon as the subject recognizes a pulse is missing at the expected pulse arrival time (Time 0), while such a program could be absent for “missing pulse” imagery because avoidance could be programmed instead. In contrast, both erroneous tapping and successful avoidance responses commonly manifested oP2 and oP3. However, this common manifestation was considered to be reasonable, taking into account that these ERP components correspond to expectation-associated internal attention driven by higher-order cognitive processing [12].

Interestingly, the omission responses for missing pulses provided almost the same onset timing as mentioned above. Furthermore, the response for erroneous tapping exhibited a reverse trace during tapping for real pulses. This suggests that such omission responses could reflect cognitive processing for late evaluation preceding imagery-driven behaviors compared with ongoing perceptual signals and modifying internal imagery models rather than representing tight connections with imagery [148]. Hence, our results and interpretations may provide a new avenue for imagery studies based on the reverse hierarchy theory. For future studies, our approach utilizing the ER-DBA method overcoming the temporal limitation imposed on the fMRI method might be useful.

## 5. Conclusions

The current study had two significant results. First, ER-DBA traces reflecting dynamic activities of the dACC may represent the contextual mental imagery that dominates behavioral responses during cued tapping, supported by the omission responses depicting reverse ERP traces during cued responses. Second, they suggest that mental imagery is pulse-by-pulse maintained by behavior-based abduction (retroduction) evaluating imagery selection errors. This maintenance is crucial for showing appropriate behavioral responses for deviant events such as a missing pulse, indicating the existence of a simple cognitive control mechanism that is similar to but different from Bayesian inference. Overall, our results suggest that the fast cognitive behavioral control by the dACC in a feedforward paradigm may be crucial for human activities based on sensorimotor synchronization, such as music playing. Considering the omission responses reflecting exogenous ERP components as a marker of sensory gating, mental imagery regulated by the dACC might cover conscious and unconscious contexts. Unconscious imagery, reflecting multimodality integration, may decrease cognitive costs. To explore the mysterious effects of music in a wide variety of fields, including rehabilitation [149,150] and psychiatric intervention [151,152], future studies will be needed to discuss this important issue.

We also highlighted role of the dACC in imagery-based cognitive control, suggesting that dACC dysfunction could result in neurological and psychiatric diseases [153,154,155,156,157]. We expect that our findings will contribute to relief of such imagery-associated symptoms whereas their radial cure methods are still developing.

## Figures and Tables

**Figure 1 brainsci-14-00757-f001:**
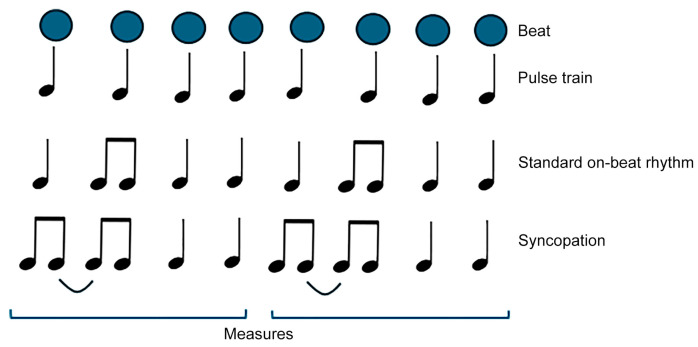
Examples of beat and rhythms.

**Figure 2 brainsci-14-00757-f002:**
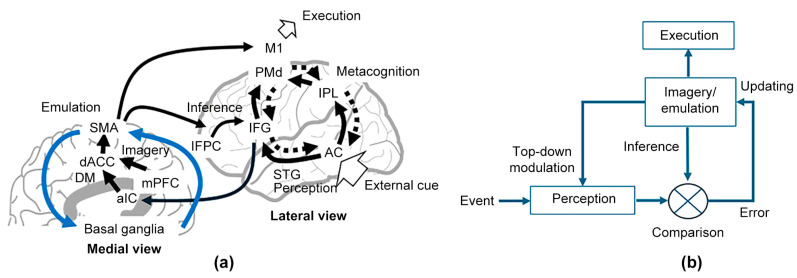
Examples of beats and rhythms. (**a**) A neural network model of imagery-driven cognitive control over rhythmic sensorimotor synchronization. (**b**) A direct imagery-execution scheme in a feedforward framework for rapid cognitive processing without Bayesian calculation.

**Figure 3 brainsci-14-00757-f003:**
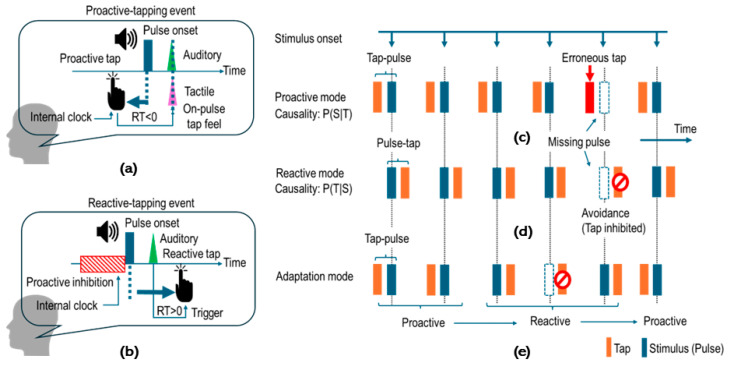
Missing-oddball task for testing the direct imagery-execution scheme. (**a**) Mental imagery of a proactive-tapping event characterized by negative mean asynchrony (reaction time (RT) < 0, NMA). NMA is attributed to auditory–tactile integration for coincidence in the brain, providing on-pulse tap feel. (**b**) Mental imagery of a reactive-tapping event characterized by positive mean asynchrony (RT > 0, PMA). The PMA is characterized by an event-triggered motor outcome regarded as reactive motor control while proactively inhibiting motor outcome until the predicted pulse-onset timing. This reactive tapping will be postponed if the pulse is missing. (**c**–**e**) are examples of postulated tapping manners in the missing-pulse sequence. (**c**) A case of tapping promoted by imagery of the proactive-tapping event. Erroneous tapping for missing pulses is unavoidable due to proactive tapping before the expected pulse-onset timing. (**d**) A case of tapping promoted by imagery of the reactive-tapping event. Erroneous tapping is avoidable. (**e**) A case of tapping promoted by appropriately exchanging imageries of proactive and reactive-tapping events, enabling proactive tapping accompanied by on-pulse tap feel while avoiding erroneous tapping for missing pulses.

**Figure 4 brainsci-14-00757-f004:**
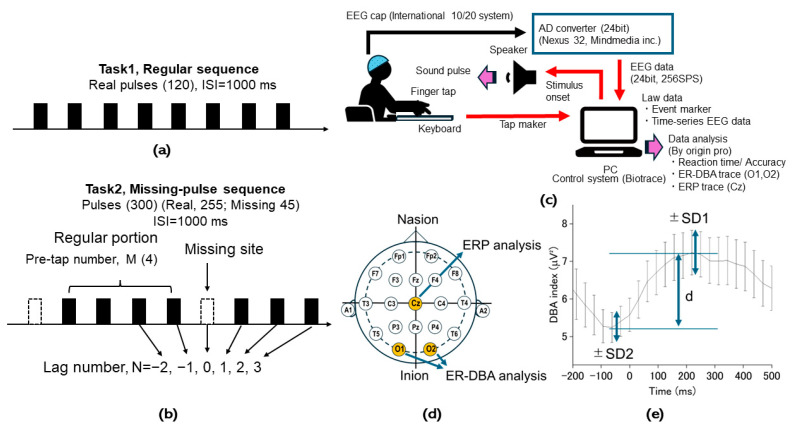
Experimental outline. (**a**) An example regular sequence consisting of 120 real pulses with an equal interval of 1000 ms used for Task 1. (**b**) An example missing-pulse sequence created by randomly omitting real pulses from the regular sequence, resulting in a total 300-pulse sequence including 45 missing and 255 real pulses. The sequence was featured by the lag number (N) locked to each missing pulse site (N = 0) and the pre-tap number (M) defined as the number of real pulses between neighboring missing sites. All real pulses had a pure tone of 1000 Hz with a duration of 100 ms. (**c**) Experimental setup. Participants were asked to simultaneously press a key with sound pulses with their eyes closed. The pulses were sequentially presented by a PC depending on the task. A digital EEG system (Nexus 32, Mind Media BV, Herten, Netherlands), comprising an EEG cap with preassembled Ag/AgCl electrodes (< 5 kΩ), consistent with the international 10–20 method, was used for EEG recordings. To integrate EEG data and event markers corresponding to tap timing, data (sampling rate: 256 Hz, amplitude resolution: 24 bit) were acquired by a PC running control software (Bio Trace+ Software for Nexus 32 Version: V2009a4). (**d**) Electrode placement of a 21-channel EEG system according to the 10–20 international system. O1 and O2 were utilized for evaluating the deep-brain activity (DBA) index while Cz was utilized for evaluating ERPs. (**e**) A sample ER-DBA trace during isochronous tapping. The trace provided a typical DBA excursion amplitude d and standard deviations at the peak and bottom, SD1 and SD2, respectively. These parameters were utilized for determining an appropriate sample size.

**Figure 5 brainsci-14-00757-f005:**
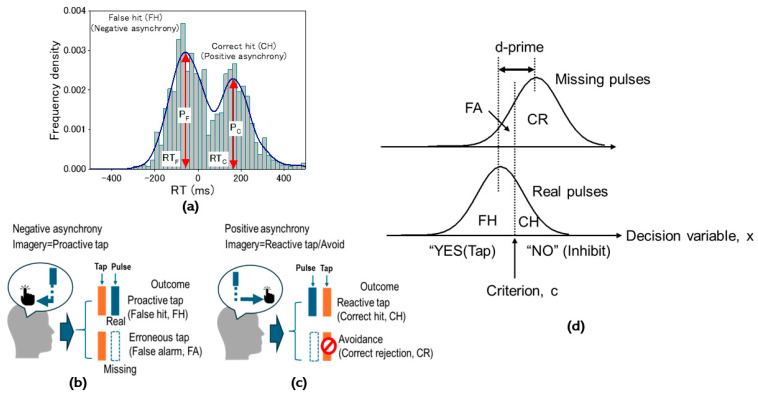
Aspects of behavioral performance during Task 2 (missing-oddball task). (**a**) Sample RT distribution for real pulses in the missing-pulse sequence resulting in two populations identified by negative and positive mean asynchronies, corresponding to proactive and reactive taps, respectively. (**b**) Proactive-tap imagery showing proactive tap outcomes as a false hit (FH) for real pulses and erroneous tap as a false alarm (FA) for missing pulses. (**c**) Reactive-tap imager showing reactive taps as a correct hit (CH) for real pulses and tap avoidance as a correct rejection (CR) for missing pulses. (**d**) Signal detection theory for the missing-oddball task indicating the probability distributions corresponding to real and missing pulses. $P_{M}\left(x\right)$ and $P_{R}\left(x\right)$ are probability density functions for “missing pulse” and “real pulse” imageries, respectively. The decision variable (x) was defined as an indicator for “Pulses are absent.” The criterion c, as an indicator of decision bias, regulates behaviors such that a tap is inhibited for x > c while being promoted for x < c.

**Figure 6 brainsci-14-00757-f006:**
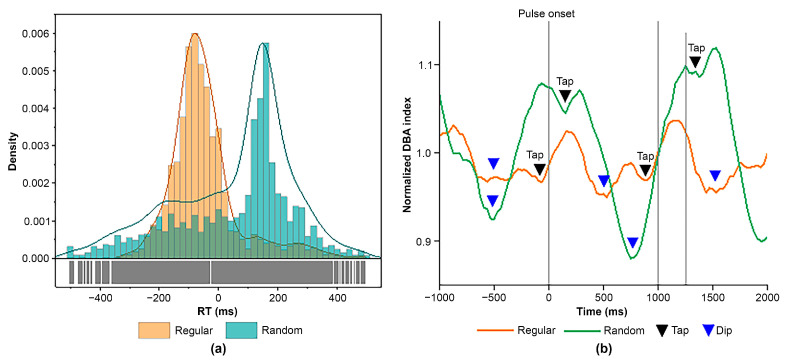
(**a**) Histogram of regular and random sequences. (**b**) Sample normalized ER-DBA traces for synchronous tapping with sequences involving regular (ISI = 1000 ms) and random (Random: 1000–1500 ms) sequences. All waveforms showed dips corresponding to each tapping timing. ISI: Inter-Stimulus Interval.

**Figure 7 brainsci-14-00757-f007:**
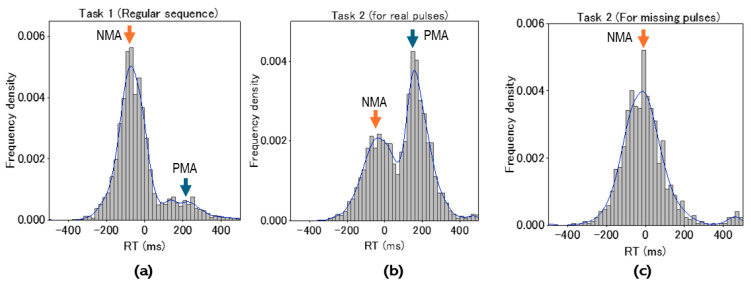
(**a**) Histogram of RTs for real pulses in the regular sequence (Task 1). (**b**) Histogram of RTs for real pulses in the missing-pulse sequence (Task 2). (**c**) Histograms of RTs for missing pulses in the missing-pulse sequence (Task 2). Distribution curves depicted in (**a**–**c**) were derived by kernel density estimation. NMA, Negative mean asynchrony; PMA, Positive mean asynchrony.

**Figure 8 brainsci-14-00757-f008:**
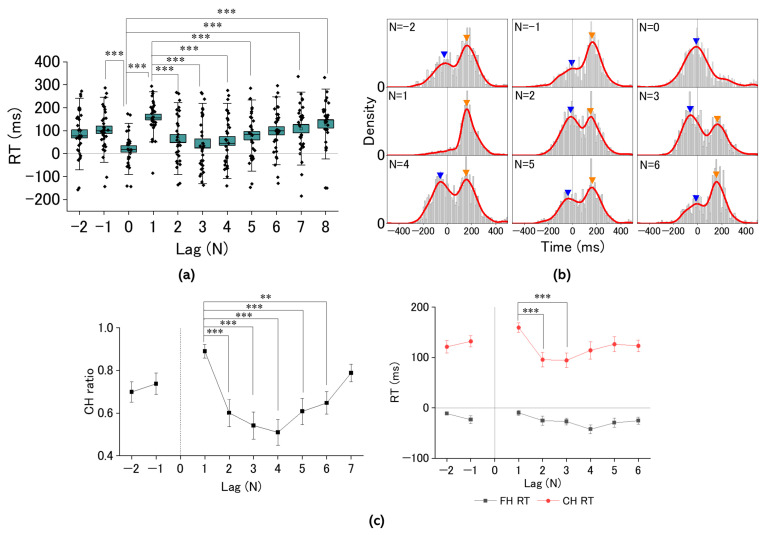
(**a**) Box-and-whisker plots of reaction times of tapping in the missing-pulse sequence (Task 2) as a function of lag number (N) locked to the missing site (N = 0). (**b**) RT histograms are represented for each lag number in the range from −2 to 6. Blue and orange triangles represent proactive-dominant and reactive-dominant asynchrony taps. (**c**) Correct-hit ratio (CHR) and reaction times of correct-hit (CH RT) and false-hit responses (FH RT) as a function of lag number (N) locked to the missing site (N = 0) derived from the individualized lag-dependent histograms in (**b**). Statistical significance is shown for comparison of RTs (**a**), CHR and CHR-associated RTs (**c**) between trials with different lag numbers and evaluated using ordinary one-way ANOVA and Holm–Sidak multiple comparisons test. (See Table 1 for numerical data). **: *p* < 0.01, ***: *p* < 0.001.

**Figure 9 brainsci-14-00757-f009:**
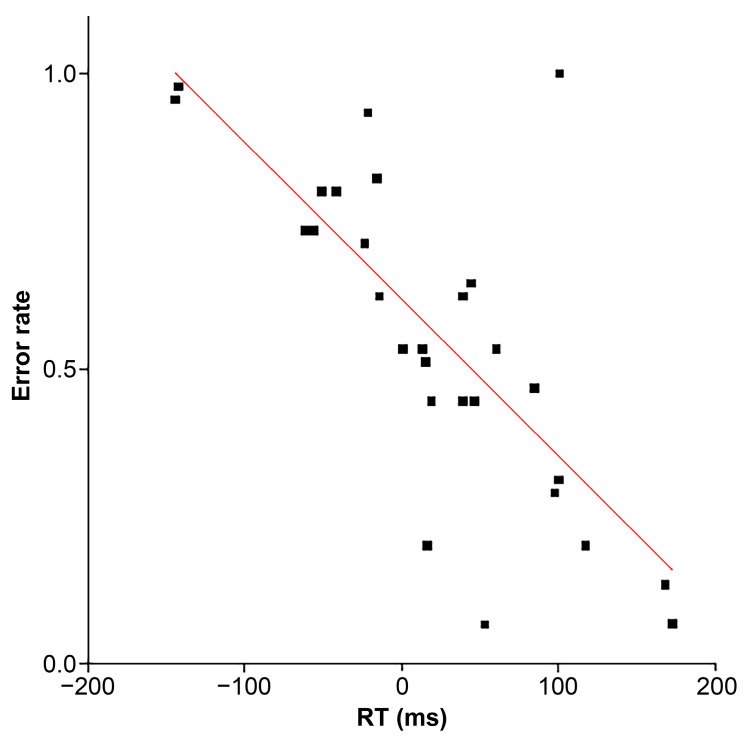
Error rates (erroneous tapping frequency for missing pulses) as a function of corresponding response times (r = −0.737, F = 33.4, h2 = 0.544, *p* = 3.32 × 10^−6^).

**Figure 10 brainsci-14-00757-f010:**
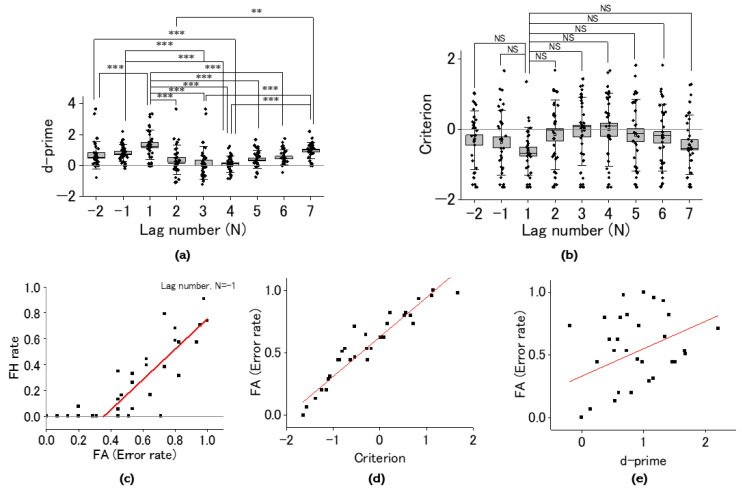
Analysis utilizing signal detection theory. Lag-number (N) dependence of d-prime (**a**) and criterion (**b**). Statistical significance is shown for comparison of average SDT parameters (d-prime (**a**) and criterion (**b**)) and evaluated using ordinary one-way ANOVA and Holm–Sidak multiple comparisons test. (See Table 2 for numerical data). (**c**) Scatter plots of R(FH) versus R(FA). R(FH) and R(FA) correspond to ratios of false hits for real pulses and FA for missing pulses of the missing-pulse sequence (Task B), respectively. R(FH) vs R(FA) > 0.4 depicts a positive linear correlation (r = 0.795, F = 36.24, *p* = 5.64 × 10^−6^, η^2^ = 0.749). (**d**) Scatter plots for the error rate of taps for missing pulses versus c depicting a strong positive linear correlation (r = 0.86, F = 80.7, *p* = 1.3 × 10^−9^, η^2^ = 0.75). (**e**) Scatter plots for the error rate of taps for missing pulses versus d-prime depicting a weak positive linear correlation (Pearson’s coefficient of correlation r = 0.41, F = 6.25, *p* = 0.0178, η^2^ = 0.167). **: *p* < 0.01, ***: *p* < 0.001, NS: *p* > 0.05.

**Figure 11 brainsci-14-00757-f011:**
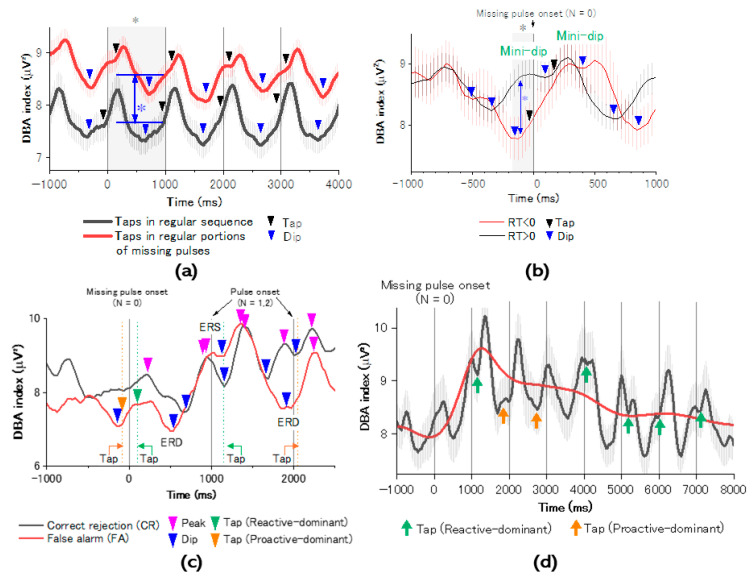
Brain responses during tasks evaluated by the ER-DBA method. (**a**) Grand-averaged ER-DBA traces for real pulses in regular (Task 1) and regular portions of the missing-pulse (Task 2) sequences. (**b**) Difference in grand-averaged ER-DBA traces between faster (RT < 0) and slower (RT > 0) responses. Dips in traces are marked by colored triangles (blue triangles for faster and slower responses, respectively). Black triangles correspond to taps. Faster and slower responses are characterized by deactivation (ERD) and activation (ERS), supported by the significant power difference in the epoch indicated by the gray area (*p* = 0.036, Power = 0.55). (**c**) Grand-averaged ER-DBA traces time-locked to missing-pulse onset compared between erroneous tapping (Tap) and CR (Avoid). Pink triangles represent ER-DBA peaks, while blue triangles represent ER-DBA dips accompanied by negative and positive asynchrony taps, respectively. (**d**) Grand-averaged ER-DBA traces at around the missing-pulse site. Orange and green arrows represent proactive-dominant and reactive-dominant asynchrony taps for real pulses in Task 2, respectively. Red line represents a temporal trajectory of averaged ER-DBA indices. *: *p* < 0.05.

**Figure 12 brainsci-14-00757-f012:**
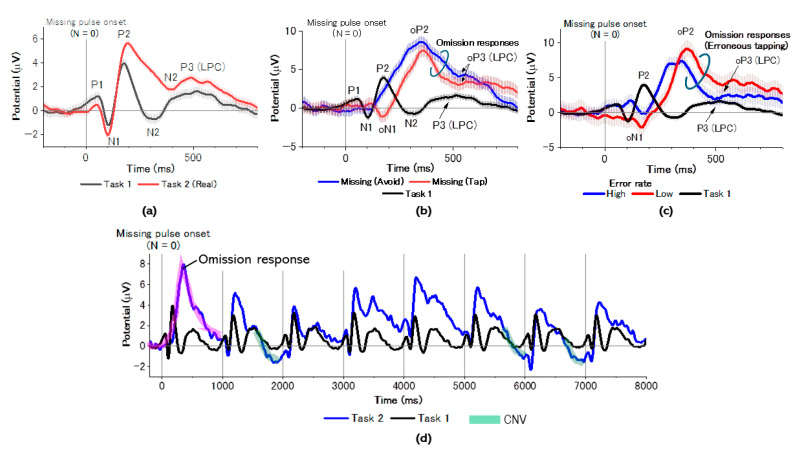
(**a**) Grand-averaged ERP traces for real pulses time-locked to stimulus onset in both regular and missing-pulse tasks excluding missing. (**b**) Grand-averaged ERP traces for missing pulses in Task 2 compared with real pulses in the same task. (**c**) ERP traces featured by erroneous tapping rate for missing pulses. Comparison among three groups, i.e., high (error rate > 0.5) and low (error rate < 0.5) error-rate trials. (**d**) Post-missing temporal development of the ERP trace. The portion of ERP trace of the highlighted by pink color is corresponds to omission responses and that highlighted by green color corresponds to CNV (contingent negative variation).

**Figure 13 brainsci-14-00757-f013:**
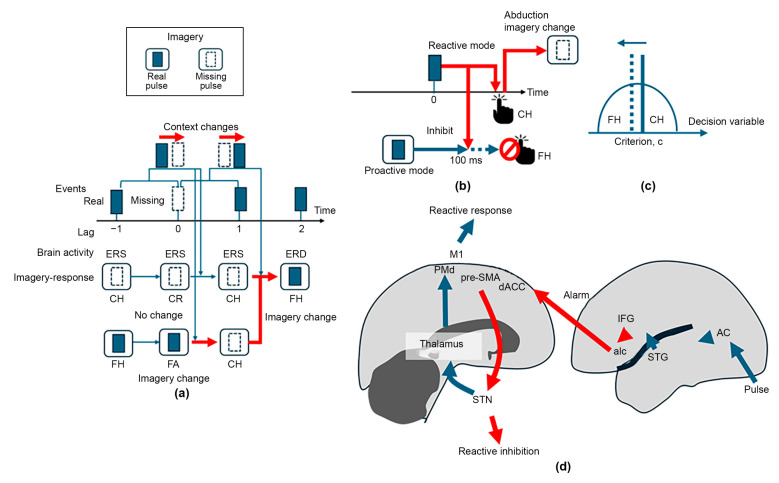
(**a**) Integration of behavioral performance, ER-DBA trace features (ERD/ERS) as brain responses, and the corresponding mental imagery based on performance and ER-DBA trace features. Imagery updating follows event changes including both “real”-to-“missing” and “missing”-to-“real” changes. (**b**) A model mechanism of imagery updates during tapping for real pulses in the missing-pulse sequence. Inhibition of delayed proactive tapping by reactive tapping triggered by real pulses may induce abduction (retroduction), where the reactive response could be attributed to the “missing” imagery. This abduction may update imagery from “real” to “missing,” while shifting the criterion toward “missing.” (**c**) A neural model to explain dominant reactive tapping for real pulses while inhibiting delayed proactive tapping. The pre-SMA stimulates the subthalamic nucleus to inhibit proactive motor outcomes via the hyper-direct pathway triggered by an alarm signal from the inferior frontal gyrus (IFG) to the dACC via the anterior insula (aIC). (**d**) A neural network model of reactive response triggered by pulse stimulation while inhibiting proactive behavior by dACC via hyper-direct pathway.

**Table 1 brainsci-14-00757-t001:** Comparison of average reaction time (RT), CHR, and CHR-associated RTs between trials with different lag numbers.

**Lag1**	**RT (ms)**	**Lag2**	**RT (ms)**	**Δ RT (ms)**	***p* Value ** **(Significant Level)**
0	20.3	−2	86.0	−65.7	0.009 NS
		−1	103.7	−83.4	0.001 ***
		1	159.8	−139.5	<0.0001 ***
		2	65.8	−45.5	0.07 NS
		3	44.5	−24.3	0.33 NS
		4	55.3	−35.0	0.16 NS
		5	78.4	−58.1	0.02 NS
		6	99.7	−79.4	0.001 ***
		7	108.8	−88.5	4.97761 × 10^−4^ ***
		8	129.3	−109.0	<0.0001 ***
1	159.8	−2	20.3	139.5	<0.0001 ***
		−1	103.7	56.1	0.001 NS
		0	20.3	139.5	<0.0001 ***
		2	65.8	94.0	0.0001 ***
		3	44.5	115.3	<0.0001 ***
		4	55.3	104.5	<0.0001 ***
		5	78.4	81.4	0.001 NS
		6	99.7	60.1	0.001 NS
		7	108.8	51.0	0.001 ***
		8	129.3	30.5	0.002 NS
**Lag1**	**CH Ratio**	**Lag2**	**CH Ratio**	**Δ CH Ratio**	***p* Value** **(Significant Level)**
1	0.9	−2	0.69898	0.2	0.00244 NS
		−1	0.73711	0.2	0.00285 NS
		2	0.60	0.3	0.000242313 ***
		3	0.54	0.3	<0.0001 ***
		4	0.51	0.4	<0.0001 ***
		5	0.61	0.3	0.000339921 ***
		6	0.65	0.2	0.00199 **
**Lag1**	**CH RT**	**Lag2**	**CH RT**	**Δ CH RT**	***p* Value** **(Significant Level)**
1	159.4	−2	121.2	38.2	0.00205 NS
		−1	132.0	27.4	0.00301 NS
		2	95.52	63.9	0.000677132 ***
		3	94.30	65.1	0.000479505 ***
		4	114.2	45.2	0.00465 NS
		5	126.4	33.1	0.0024 NS
		6	94.30	65.1	0.000479505 ***
**Lag1**	**FH RT**	**Lag2**	**FH RT**	**Δ FH RT**	***p* Value** **(Significant Level)**
1	−9.8	−2	−11.50	1.7	0.88 NS
		−1	−23.52	13.7	0.23 NS
		2	−25.30	15.5	0.18 NS
		3	−27.10	17.2	0.11 NS
		4	−42.51	32.7	0.06 NS
		5	−29.70	19.8	0.08 NS
		6	−25.57	15.7	0.84 NS

Lag number: 0, missing slot; −1, pre-missing slot; 1,2, —post missing slot Statistical significance was evaluated using ordinary one-way ANOVA and Holm–Sidak t-test multiple comparisons test. Significant difference in Lag1 and Lag2; **: *p* < 0.01, ***: *p* < 0.001, NS: *p* > 0.05.

**Table 2 brainsci-14-00757-t002:** Comparison of average (±SE) d-prime and criterion (c) between trials with different lag numbers (Lag1 and Lag2).

**Lag1**	**d-Prime1**	**Lag2**	**d-Prime2**	**Δ d-Prime**	***p* Value** **(Significant Level)**
−2	0.67	1	1.32	−0.65	0.00048 ***
		4	0.09	0.59	0.001 ***
−1	0.81	3	0.17	0.64	0.0006 ***
		4	0.09	0.73	0.0001 ***
1	1.32	2	0.95	0.37	<0.0001 ***
		3	1.05	−1.05	<0.0001 ***
		4	0.51	−0.51	<0.0001 ***
		5	0.56	−0.56	<0.0001 ***
		6	0.48	−0.48	<0.0001 ***
2	0.95	7	0.53	0.42	<0.0001 ***
3	0.37	7	0.53	−0.16	<0.0001 ***
4	0.51	7	0.53	−0.02	<0.0001 ***
**Lag1**	**Criterion1**	**Lag2**	**Criterion2**	**Δ Criterion**	***p* Value** **(Significant Level)**
1	−0.63	−2	−0.31	−0.33	0.15 NS
		−1	−0.37	0.37	0.25 NS
		2	−0.16	0.16	0.03 NS
		3	−0.05	0.05	0.01 NS
		4	−0.01	0.01	0.006 NS
		5	−0.17	0.17	0.04 NS
		6	−0.23	0.23	0.08 NS
		7	−0.45	0.45	0.41 NS

Lag number: 0, missing slot; −1, pre-missing slot; 1,2, ---post missing slot Statistical significance was evaluated using ordinary one-way ANOVA and Holm–Sidak *t*-test multiple comparisons test. Significant difference in Lag1 and Lag2; ***: *p* < 0.001, NS: *p* > 0.05.

## Data Availability

The datasets used and/or analyzed during the current study are available from the corresponding author on reasonable request. The data are not publicly available due to ethical restrictions.

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
