# Peer review of "Dorsal Anterior Cingulate Cortex Coordinates Contextual Mental Imagery for Single-Beat Manipulation during Rhythmic Sensorimotor Synchronization"

_brainsci, 2024, doi:10.3390/brainsci14080757_

Round 1
Reviewer 1 Report (New Reviewer)
Comments and Suggestions for Authors
The presented work provides valuable insights into the role of the dACC in rhythmic sensorimotor synchronization. By addressing different scientific issues and implementing the suggested improvements, the paper could be considered for the final publication:
1. The hypothesis regarding the role of the dorsal anterior cingulate cortex (dACC) in coordinating proactive and reactive motor outcomes based on contextual mental imagery is intriguing. However, I would suggest to highlight the research objectives within the Introduction section.
2. The performed literature review appears to be incomplete for specific aspects. Therefore, I suggest to better investigate the state-of-the-art contributions, for example, by distinguishing clearly between proactive and reactive motor control studies and their respective findings could enhance clarity.
3. The use of EEG and ERP-derived measures is appropriate, but the paper could benefit from a more detailed explanation of the preprocessing steps and criteria for artifact rejection. This is a significant aspect of the presented work.
4. There is no a reasonable justification of the selected sample size.
5. I would suggest to better describe the processing of the event-related deep-brain activity (ER-DBA), since it is quite unclear.
6. There is a lack of details on the statistical methods used to compare proactive and reactive tapping behaviors. I suggest to better describe any corrections for multiple comparisons and the exact p-values where significance was claimed.
7. I suggest to discuss potential confounding variables and their possible impact on the results. In fact, the suggested connection between ERPs and mental imagery is not robustly supported by the data.
8. I suggest to improve the cleareness of specific figures, such as the ones illustrating the experimental setup and results.
Author Response
Please see the attachment

Reviewer 2 Report (New Reviewer)
Comments and Suggestions for Authors
“Dorsal anterior cingulate cortex coordinates contextual mental imagery for single-beat manipulation during rhythmic sensorimotor synchronization”( brainsci-3084225)
This manuscript aimed to determine whether the dorsal anterior cingulate cortex (dACC) plays a key role by coordinating both proactive and reactive motor outcomes based on contextual mental imagery. A missing-oddball task in finger-tapping paradigms was conducted in 33 healthy young volunteers. The dynamic properties of the dACC were evaluated by event-related deep-brain activity (ER-DBA), supported by event-related potential (ERP) analysis and behavioral evaluation based on signal detection theory. The results revealed that ER-DBA activation/deactivation reflected a strategic choice of motor control modality in accordance with mental imagery. Reverse ERP traces, as omission responses, confirmed that the imagery was contextual. We found that mental imagery was updated only by environmental changes via perceptual evidence and response-based abductive reasoning. Moreover, stable on-pulse tapping was achievable by maintaining proactive control while creating an imagery of syncopated rhythms from simple beat trains, whereas accuracy was degraded with frequent erroneous tapping for missing pulses. These results suggested that dACC voluntarily regulates rhythmic sensorimotor synchronization by utilizing contextual mental imagery based on experience and by creating novel rhythms. Overall, this topic is interesting and the current investigation provides some novel and vital insights about the neuroscience mechanism during rhythmic sensorimotor synchronization. Moreover, the usage of event-related deep-brain activity and signal detection theory also provide some methodological implications for future studies. The manuscript is well-written, organized and the literature referred is comprehensive and up-dated and I enjoyed reading the whole manuscript. There are only some minor concerns which need to be clarified.
1. How did you determine the sample size? Did you calculate the sample size needed before formal study?
2. For the EEG cap, how many electrodes in total in the cap and did you use all the electrodes or not?A figure to depict all the electrodes you used might be helpful to the readers.
3. What tool did you use for the ERP analysis?
4. A separate paragraph about the limitations and future directions at the end of discussion part might be helpful for the readers and inspire future studies.
Comments on the Quality of English Language
Minor editing of English language required
Round 2
Reviewer 1 Report (New Reviewer)
Comments and Suggestions for Authors
Thank you to the Authors for the updated version of the manuscript. All the issues were sufficiently addressed.
This manuscript is a resubmission of an earlier submission. The following is a list of the peer review reports and author responses from that submission.
Round 1
Reviewer 1 Report
Comments and Suggestions for Authors
The paper's premise is intriguing and holds significance. The methods employed are appropriately applied, and the data effectively substantiate the conclusions drawn. The writing flows smoothly, and the English composition is commendable. Nevertheless, I have two minor concerns that could enhance the overall robustness of the study:
1. Could you provide clarity on the number of brain regions recorded during the task? Is the dACC the sole region demonstrating a correlation with sensorimotor synchronization, or are there additional regions involved? It would strengthen the argument if other brain regions were identified to underscore the specificity of the dACC in relation to sensorimotor synchronization.
2. Are there any existing studies detailing the symptoms observed in patients with dACC damage? If such literature exists, could the symptoms observed in these patients find a reasonable explanation in the context of your research? If available, please incorporate these references into your manuscript, as they would significantly enrich the discussion and overall impact of the paper.
Comments on the Quality of English Language
The writing flows smoothly, and the English composition is commendable.
Reviewer 2 Report
Comments and Suggestions for Authors
The study's hypothesis is interesting, but the execution and presentation of the research do not meet the standards required for publication. Here are my comments:
- The introduction, particularly the paragraph between lines 61 and 82, lacks clarity. This section is crucial as it provides the rationale for the study, but its current form is very hard to follow.
- The decision to include only female participants is not adequately justified. The acknowledged presence of sex differences does not justify excluding male subjects.
- There are numerous instances where the instructions from the journal are still present in the manuscript (e.g., lines 133-147, 263-265). This suggests a lack of careful review and preparation of the manuscript before submission.
- Figure 2(a) and (c): In the case of P-musicians, it's unclear why all mean reaction times (RTs) for real pulses are positive in Figure 2(a), yet Figure 2(c) shows a significant negative peak.
- Figure 2 legend: what are S- and B- musicians?
- Line 297 - 308: how exactly was the statistics test done? Why do you call it "group difference" but you compared ERP to the baseline (two p-values, one for each musician type)? What is the N? Why are the Ns so large?
- In general, separating the data into P-musicians and N-musicians and then applying analyses to each group separately seems methodologically questionable, resembling a form of 'double-dipping'. This concern is compounded by the apparently small sample size (N=14).
- The discussion surrounding free-energy theory seems to be an overstretch, potentially leading to overinterpretation of the data.
Comments on the Quality of English Language
Extensive editing of English language required.
Reviewer 3 Report
Comments and Suggestions for Authors
The manuscript is well structured and deals with a topical topic of potential interest for the scientific community. However, I have some suggestions for authors. In the "Introduction" paragraph, in the final part, the purpose of the study should be clearly specified so as to make it clear for readers. Additionally, to lightly implement this section, they might consider the following recent scientific work: - Moscatelli et al., High frequencies (HF) repetitive transcranial magnetic stimulation (rTMS) increase motor coordination performances in volleyball players, BMC Neurosciences, 2023, 24(1), 30.